# Structure of human Dispatched-1 provides insights into Hedgehog ligand biogenesis

Hongwen Chen[1,*] , Yang Liu[1,*] , Xiaochun Li[1,2]

**Hedgehog (HH) signaling is essential for metazoan development. The HH ligand is secreted into the extracellular space by a cell surface protein named Dispatched-1 (DISP1). Here, we report the cryo-EM structure of human DISP1 protein. DISP1 contains 12 transmembrane helices (TMs) and two extracellular domains (ECDs). Its ECDs reveal an open state, in contrast to its structural homologues PTCH1 and NPC1, whose extracellular/luminal domains adopt a closed state. The low-resolution structure of the DISP1 complex with dual lipid-modified HH ligand reveals how the ECDs of DISP1 engage with HH ligand. Moreover, several cholesterol-like molecules are found in the TMs, implying a transport-like function of DISP1.**

## Introduction

Hedgehog signaling is essential for embryonic development, whereas aberrant signaling is implicated in various cancers (Briscoe & Therond, 2013; Pak & Segal, 2016; Hu & Song, 2019; Qi & Li, 2020). HH protein is one of many morphogens that control the organization of cells into tissues during embryogenesis (Hall et al, 2019). The full-length HH protein is cleaved by itself into two fragments: HH-N and HH-C (Lee et al, 1994). The HH-C can covalently transfer a cholesterol molecule to the C terminus of HH-N (Porter et al, 1996). Subsequently, a Hedgehog acyltransferase (HHAT) transfers a palmitate moiety to the N terminus of HH-N (Chamoun et al, 2001). The dual lipid-modified HH-N (referred to as "native HH-N") is transported to the cell surface by an unknown mechanism. At the plasma membrane, HH-N is first bound by DISP1 and then released into the extracellular space by a molecular chaperone named SCUBE2 (signal peptide, CUB, and EGF-like domain-containing protein 2). Notably, dual lipid modifications play essential roles to form the complex with HH-N binders: 1) the cholesterol modification is indispensable for HH secretion into the extracellular space by DISP1 and SCUBE2 (Creanga et al, 2012; Tukachinsky et al, 2012) and 2) the palmitate of HH-N binds to

PTCH1 to block the sterol tunnel in PTCH1 triggering the HH signal (Zhang et al, 2018; Qi et al, 2018a, 2018b, 2019b).

The cholesterol modification of HH is important for generating the gradient of the HH morphogen in tissues during embryogenesis (Gallet et al, 2003). In mice, knockout of the *DISP1* homolog *Disp1* exhibits defective development on the neural tube, suggesting an important role of Disp1 in the formation of this gradient (Ma et al, 2002). In humans, there are three DISP isoforms. DISP1 is a well-studied isoform that contains 1,524 amino acid residues, including a ~170-residue N-terminal intracellular domain (NTD), 12 transmembrane helices (TMs), two extracellular domains (ECDs), and a ~300-residue C-terminal intracellular domain (CTD) (Fig 1A). As a member of the resistance–nodulation–division (RND) transporters, DISP1 shares a similar topology with PTCH1, NPC1, and bacterial efflux transporters such as AcrB (Scott & Ioannou, 2004; Li et al, 2016b). TMs 2-6 of DISP1 form the sterol-sensing domain (SSD), which is widely conserved among polytopic membrane proteins involved in cholesterol metabolism and transport (e.g., PTCH1 and NPC1) (Goldstein et al, 2006). Previous studies have revealed that PTCH1 (Zhang et al, 2018; Qi et al, 2018a) and NPC1 (Winkler et al, 2019; Long et al, 2020) transport cholesterol using an internal channel, whereas the SSD serves as a gate to the tunnel for cholesterol trafficking through the respective protein; the similarity of these two transporters is further underscored by related inhibitory mechanisms (Long et al, 2020). Finally, alteration of three Asp residues in the TMs of DISP1, which are putatively involved in the protein's transporter-like function, affects DISP1 activity (Ma et al, 2002; Tukachinsky et al, 2012). Despite these extensive functional studies, the mechanism of DISP1-mediated transport activity remains unknown.

Recently, a remarkable study showed that Furin, a proprotein convertase, can cleave the loop of DISP1-ECD-I (between residues Lys279 and Arg280 of human DISP1), which promotes HH-N release and is in fact required for the function of DISP1 in vivo (Stewart et al, 2018). Although this cleavage does not interfere with the interactions between HH-N and DISP1, mutations of the cleavage site block HH release. Notably, such a cleavage has not been observed in the DISP1 structural homologues PTCH1 and NPC1. However, the

[1]Department of Molecular Genetics, University of Texas Southwestern Medical Center, Dallas, TX, USA   [2]Department of Biophysics, University of Texas Southwestern Medical Center, Dallas, TX, USA

Correspondence: xiaochun.li@utsouthwestern.edu
*Hongwen Chen and Yang Liu contributed equally to this work

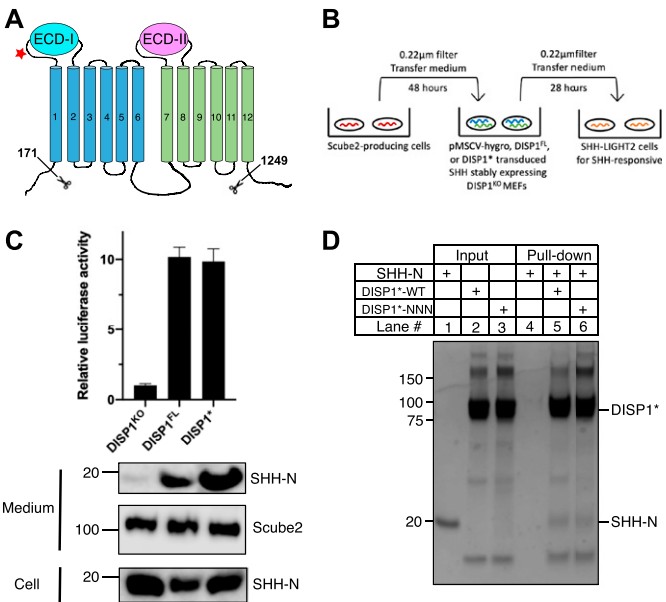

**Figure 1.   Functional validation of DISP1\* for structural investigation.**
**(A)** A diagram of topology of human DISP1. Residues 1–171 and 1,249–1,524 were removed in DISP1\*. **(B)** Experimental scheme to measure SHH release capacity. HEK293 Flp-In T-REX cells were stably transfected with an inducible 3× Flag-tagged Scube2. Cells were induced and the supernatant containing Scube2 conditioned medium was collected, filtered, and mixed with fresh medium at a 1:1 ratio and finally transferred to DISP1$^{-/-}$ or DISP1\* and DISP$^{FL}$ rescued MEF cells stably expressing FL-SHH. After 48 h of incubation, the SHH-N conditioned medium was harvested and mixed with fresh medium at 1:1 ratio to incubate with SHH-Light II cells. **(C)** SHH-N release in DISP1$^{-/-}$ MEF cells that were transduced with both SHH and empty vector (DISP1$^{KO}$), full-length DISP1 (DISP1$^{FL}$) or DISP1\* was checked via dual luciferase assay where the SHH-Light II cells stably express firefly luciferase with an 8×-Gli promoter and *Renilla* luciferase with a constitutive promoter. Data are mean ± SD (n = 8 biologically independent experiments). **(D)** Pull-down assay of native SHH-N with DISP1\*-WT or DISP1\*–NNN mutant detected by Coomassie staining.

mechanism underlying DISP1-mediated HH-N release from the morphogen-producing cell remains to be elucidated. We here report the cryo-EM structures of human DISP1 and its complex with dual lipid-modified HH ligand, which reveal an open conformation adopted by ECDs for HH binding and will extend our current understanding of HH-N biogenesis.

## Results and Discussion

The expression of full-length human DISP1 (hDISP1) in human embryonic kidney (HEK) cells turned out to be insufficient for structural studies. Secondary structure predictions suggest that the NTD and CTD of hDISP1 are flexible, and sequence alignment demonstrated that these two regions are not conserved among various species. Therefore, we screened human hDISP1 constructs with different N- and C-terminal truncations. The construct referred to as DISP1\* with an N-terminal truncation (Δ1-171) and a C-terminal truncation (Δ1249-1524) yielded high expression in HEK cells with ideal behavior in a detergent solution (Fig S1A).

Studies on the DISP1-associated N-terminal fragment of Sonic Hedgehog (SHH-N) release revealed that the secretion of SHH-N

requires a glycoprotein, SCUBE2, and the C-terminal cholesterol moiety on SHH-N (Tukachinsky et al, 2012). To verify if DISP1\* functionally retained the ability to bind and secrete SHH-N into the extracellular milieu, we first transduced *Disp1*-deficient MEFs with full-length SHH (FL-SHH) (Fig 1B). As expected, cellular FL-SHH was processed to generate SHH-N but failed to exit the cell even in the presence of SCUBE2 (Fig 1C, bottom). We then rescued expression of DISP1 with DISP1\* or full-length DISP1 (DISP1$^{FL}$) and could detect processed SHH-N in both lines. Furthermore, the secreted SHH-N functionally activated SHH-Light II cells, an NIH 3T3-derived line stably transfected with Gli-dependent firefly luciferase and constitutive *Renilla* luciferase. Notably, qRT-PCR (Fig S2) showed an approximately threefold higher expression of DISP1\* than DISP1$^{FL}$. Despite this higher expression, the overall amount of secreted SHH-N was relatively similar. This is consistent with a previous report showing that DISP1 lacking the C-terminal domain results in a mild reduction of SHH secretion (Etheridge et al, 2010). Even so, these results show that DISP1\* is functional and suitable for further studies.

Next, our in vitro binding assays confirm that DISP1\* can bind native SHH-N ligand in a detergent solution (Fig 1D). For structural investigations by cryo-EM, we mixed DISP1\* with native SHH-N protein at a 1:1.1 molar ratio and purified the complex by size-exclusion chromatography in a buffer containing 0.06% digitonin and 0.002% CHS (Fig S1B). The particles of the DISP1\*–SHH-N complex appear homogeneous (Fig S3A). However, after 3D classification, the cryo-EM map of SHH-N in the complex appeared weak (Fig S4A). It is possible that either the DISP1\*–SHH-N complex disassociates during grid preparation or that SHH-N adopts multiple conformations upon binding DISP1\*, thus leading to a weak cryo-EM map of SHH-N after 3D classification. Further 3D classifications were performed to sort the particles for final refinement (Figs S3A and S4A).

The structure of DISP1\* was determined at 4.5-Å resolution (Figs 2A–C, S3B, and Table S1); because of the higher local resolution, the TMs are well modeled and most aromatic residues could be assigned (Fig S5). Because the DISP1\* protein appears as two bands on an SDS–PAGE (Fig S1A) and the loop region 253–292 is invisible in the cryo-EM map (Fig 2A and B), we speculated that the recombinant DISP1\* was cleaved by Furin, which is consistent with a previous study showing that Furin-mediated cleavage is required for DISP1 maturation and localization (Stewart et al, 2018). Because of weak density or limited local resolution of ECDs and some linkers, some residues in these regions have not been built or built as polyalanine (also see the Materials and Methods section), suggesting high flexibility within these regions.

The overall dimensions of DISP1\* are 60\*85\*90 Å (Fig 2B and C), shorter and wider than its structural homologues PTCH1 (pdb:6OEU) and NPC1 (pdb:5U74, without the N-terminal domain and NPC1-TM1 that do not exist in PTCH1 or DISP1) (Fig 3A and B). Structural comparison revealed an overall root-mean-square deviation (RMSD) of 7 Å between DISP1\* and PTCH1 and an overall RMSD of 6 Å between DISP1\* and NPC1. Residues 757–776 form a short α-helix (α1\*) that closely interacts with ECD-I (residues 293–490) (Fig 2B). Similarly, residues 227–243 form a short α-helix (α1) followed by a loop region (residues 244–253), which interacts with ECD-II (residues 785–970) (Fig 2B). The ECDs of PTCH1 and NPC1 twist together to create an

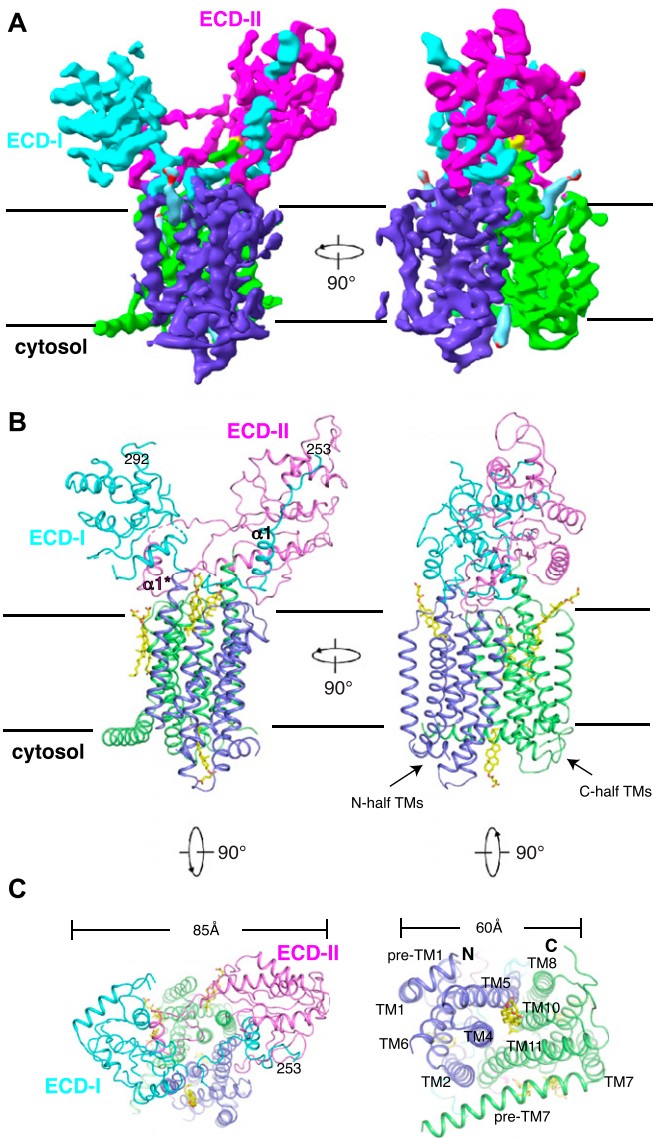

**Figure 2. Overall Structure of DISP1*.**
**(A)** Cryo-EM map after final RELION-3 refinement sharpened using "post-processing." **(B)** Overall structure showing DISP1* viewed from the side of the membrane. **(C)** Ribbon representations of the structure viewed from extracellular and cytosolic side, respectively. Residues that are closed to the Furin cleavage site, as well as transmembrane helices with the putative cholesteryl hemisuccinate (yellow sticks) are labeled.

intermolecular tunnel for cholesterol transport (Qi et al, 2018a; Long et al, 2020); in contrast, ECD-I of DISP1* is shifted from ECD-II by more than 15 Å, which is not observed in either NPC1 or PTCH1 (Fig 3B), leading to an open conformation of the two ECDs in DISP1*. The α1 helix can be well aligned with the corresponding helix of PTCH1 and NPC1 (Fig 3A and B). On the other hand, the loop between residues 253–292, which is absent in both PTCH1 and NPC1, is disordered in the map (Fig 2B and C), indicating that the flexibility of this large loop permits it to be accessible to Furin and the cleavage by Furin may release this internal loop to facilitate the endosomal trafficking of DISP1 (Stewart et al, 2018).

Previous structural studies showed that ECD-I and ECD-II of PTCH1 and the middle luminal domain of NPC1 could bind SHH-N ligand (Gong et al, 2018; Qi et al, 2018a, 2018b) and NPC2 (Li et al, 2016a), respectively. The density of SHH-N in the DISP1*–SHH-N complex has been observed at a low threshold after the 3D classification (Fig S4A). We created a mask of ECDs and SHH-N for further 3D classification, with the resulting class containing more than 110,000 particles. We polished the particles using Relion-3 and performed a refinement with a soft mask. The final resolution of the complex is 7.9 Å (Fig S4B and Table S1), but the secondary structure elements of SHH-N are still not well defined in the cryo-EM map. The reason for this observation remains unknown. As stated earlier, it is possible that either the DISP1*–SHH-N complex disassociates during grid preparation or that SHH-N binds DISP1* dynamically allowing SHH-N to shed. Although the map of SHH-N is not well resolved, we can still model it according to the extra bulky map in the extracellular space (Fig 3C). The putative SHH-N binds both ECDs of DISP1* (Fig 3D). The structure of PTCH1–SHH-N complex revealed two interfaces between SHH-N and PTCH1 (Fig 3E) (Qi et al, 2018a). First, the palmitate-dominated interface binds to ECD-I of PTCH1-A. Second, the calcium-mediated interface binds to both ECDs of PTCH1-B (Fig 3E). The structural comparison shows that the binding mode of putative SHH-N to DISP1* is similar to that of PTCH1-B whose two ECDs are involved in interactions. The SHH-N can engage two PTCH1 molecules via its distinct interfaces to form a 2:1 complex (Qi et al, 2018a). However, it remains unknown whether SHH-N can also bind two DISP1* molecules with different surface residues to form an oligomer.

Our previous studies showed that the palmitoyl moiety can insert into the center of PTCH1-A to block its intermolecular tunnel (Qi et al, 2018a, 2018b). In the putative complex of SHH-N and DISP1*, we cannot resolve how the two lipid modifications of SHH-N impact complex assembly. Previous structures demonstrate that the ECDs of PTCH1 can accommodate a sterol-like molecule (Gong et al, 2018; Zhang et al, 2018; Qi et al, 2018a, 2019a; Rudolf et al, 2019) (Fig S6). Therefore, the cholesterol modification possibly binds either ECD-I or ECD-II of DISP1* to enhance the receptor–ligand interactions.

An internal C2 symmetry is found in the TMs of DISP1* (Fig 2C). TMs 1-6 (N-half TMs) share a conformation similar to TMs 7-12 (C-half TMs) with an overall RMSD of 2.2 Å. The N terminus of TM2 is shifted more than 6 Å towards the edge of TMs compared with TM2 of PTCH1 (Fig 4A); this shift is presumably caused by the open conformation of ECD-I because TM2 directly connects to ECD-I (Fig 3A). As a result of this shift, the N terminus of TM4 and TM11 reveal an ~4–5 Å shift compared with that of PTCH1, avoiding a steric clash with TM2 (Fig 4A).

Similar to the sterol-like molecules found in the TMs of PTCH1 (Qi et al, 2018b, 2019a), five sterol-like molecules are observed in the TMs of DISP1 (Fig 4B). We tentatively assign these molecules as cholesteryl hemisuccinate (CHS) based on the density shapes and the hydrophobic environment that is created by potential binding residues. One is found in the sterol-binding pocket of the SSD (site 1), and two are found near the N terminus of TM12 (site 2). These findings are consistent with the positions of sterols in the transmembrane domains of PTCH1 (Fig S6). There are two novel sterol-binding sites in DISP1*: One is located in the pocket created by TMs 7-10 (site 3) in a position that is symmetrically related to the sterol

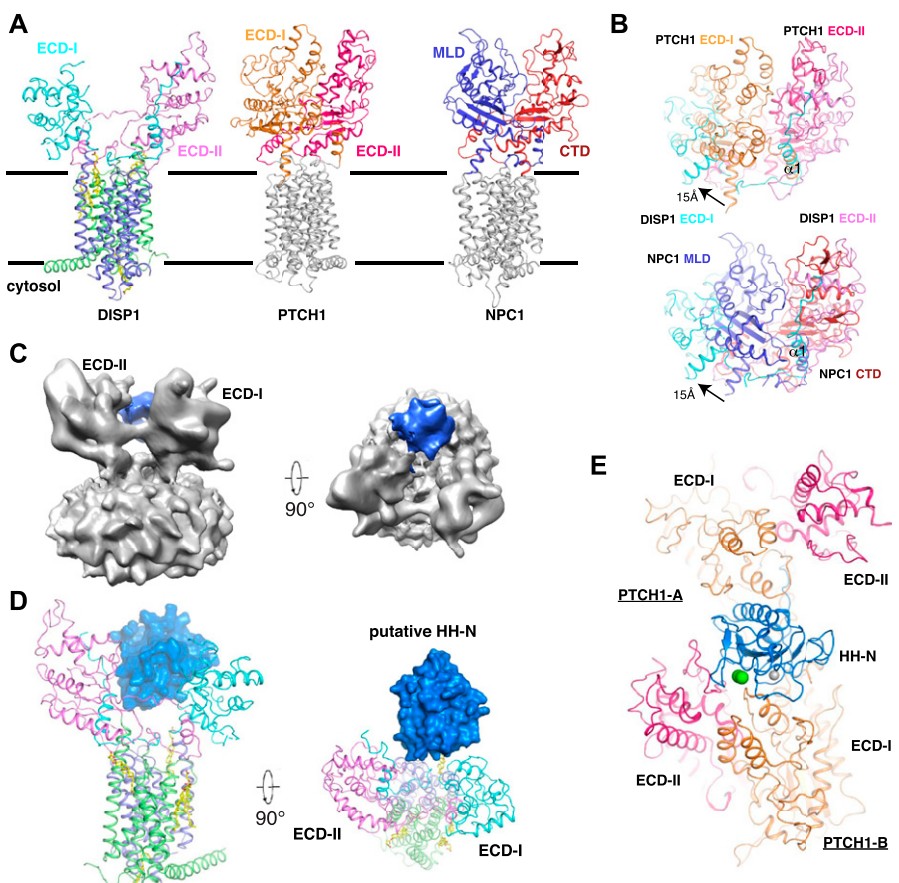

**Figure 3. Structural comparison of DISP1\* with PTCH1 and NPC1.**
**(A)** Structural comparison of DISP1\* with PTCH1 (pdb: 6OEU) and NPC1 (pdb: 5U74). **(B)** Superposition of extracellular domains (ECDs) of DISP1\* with the counterparts of PTCH1 and NPC1, respectively. A 15-Å shift of ECD-I of DISP1\* from ECD-II is labeled. **(C)** cryo-EM map of DISP1\* complex with SHH-N from Relion-3 output. **(D)** Ribbon representation of the complex structure. The putative SHH-N ligand is shown as a surface model in blue. **(E)** The structure of PTCH1 with its ligand SHH-N (pdb: 6E1H).

in the SSD. The other one is found toward the cytoplasmic side of the protein, and its binding site is created by TMs 4, 5, 10, and 11 (site 4). The sterol binding in site 4 introduces conformational changes of the C terminus of TMs 5 and 10 of 4 Å compared with that of PTCH1 (Fig 4C).

Asp572, Asp 573, and Asp1051, which are conserved in PTCH1, are located ~10 Å above the putative cholesterol-bound site 4 in TMs (Fig 4B). Although it has been well established that most RND superfamily proteins function as H⁺-driven antiporters (Tseng et al, 1999; Nikaido, 2018), some RND transporters in halophilic organisms have been identified to be Na⁺-driven (Ishii et al, 2015). Neutralization of the analogous PTCH1 residues (D499N, D500N, and E1081Q) disrupts the activity of PTCH1, thereby inhibiting its downstream signaling, which suggests that a Na⁺ gradient provides the energy source for transport activity of PTCH1 (Myers et al, 2017). The corresponding mutations in DISP1 abolish the release of HH-N in cells, but still retain binding of HH-N to DISP1 (Tukachinsky et al, 2012) (Fig 1D). Interestingly, a structural comparison shows that Asp573 and Asp1051 are 6 Å apart, which is further than the 3 Å-distance between the corresponding residues in PTCH1 or NPC1 (Fig 4D and E). This structural analysis suggests that the three Asp residues in DISP1 may constitute a core ion-conducting circuit to use a different ion gradient to provide the energy source for transporter-like activity. However, the relation between transporter-like activity and the release of HH-N is still unclear. One appealing hypothesis is

that the transmembrane ion gradient drives DISP1 to transport cholesterol or its derivatives between inner and outer leaflets, which regulates the release of HH-N. Alternatively, the cholesterol-like molecule in site 4 may increase cytoplasmic stability between the N- and C-half TMs (Fig 4C) or serve as an allosteric molecule to ensure that DISP1 is in a certain conformation to generate the HH morphogen gradient. A recent study reported that an NPC1 variant without a cytoplasmic loop connecting the N- to the C-half TMs loses its cholesterol transport activity, suggesting that the dynamic of the NPC1 helices is necessary for cholesterol transport (Saha et al, 2020).

Previous structural studies showed that either the palmitoyl moiety of HH-N or a small molecule named itraconazole can insert into the cavity of PTCH1 or NPC1 to block cholesterol transport (Qi et al, 2018a, 2018b; Long et al, 2020). Unlike PTCH1 and NPC1, the ECDs of DISP1\* adopt an open conformation and lack an interdomain hydrophobic cavity; therefore, the mechanism by which the transport activity of DISP1 is inhibited remains unknown.

While this manuscript is under preparation, the structures of *Drosophila* DISP (dDISP) and its complex with unmodified ligand HH-N have been reported (Cannac et al, 2020). The structural comparison reveals that DISP1\* shares a similar conformation to dDISP and the unmodified *Drosophila* HH-N ligand binds to two ECDs similarly to human HH-N (Figs 3C and S7). The secondary structures of unmodified *Drosophila* HH-N can be defined in the

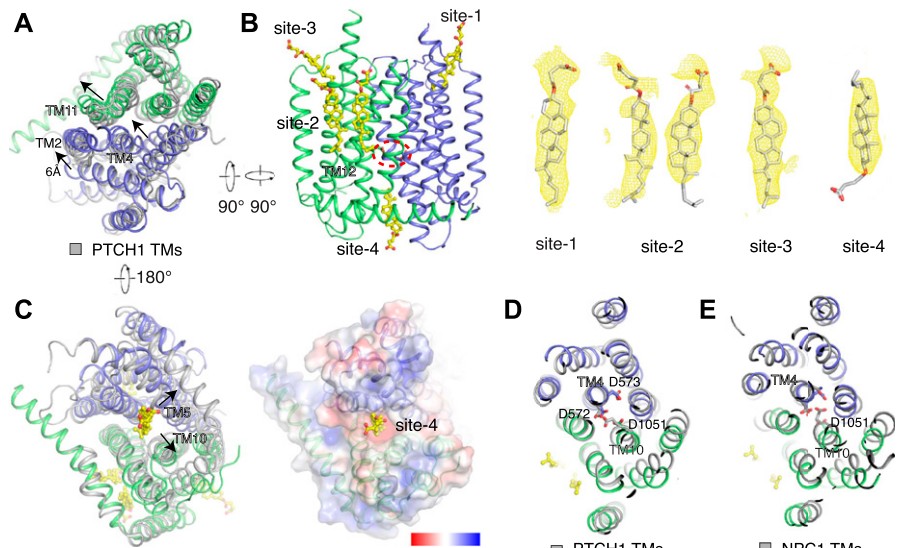

**Figure 4.   The transmembrane domain of DISP1* with sterol-like ligands.**
**(A)** Structural comparison of the transmembrane domains of DISP1* with PTCH1. **(B)** The four sterol-bound sites in the TMs of DISP1*. The location of the three Asp residues is indicated by a red dashed circle. EM maps and models of cholesteryl hemisuccinate are shown at 5σ level in mesh and cartoon. **(C)** The putative sterol in site 4 triggers the conformational changes of TMs. Electrostatic surface representation is shown in the right panel. **(D, E)** The structural comparison shows that the sterol in site 4 triggers a conformational change of TM10 to provide the cytoplasmic mobility between the N-half TMs and C-half TMs.

cryo-EM; therefore, we speculate that the cholesterol modification of HH-N may promote the shed of HH-N causing it untraceable in the cryo-EM map. The structures of dDISP1 and hDISP1 suggest an evolutionary conservation in HH-N recognition across the species. No sterol-like molecule has been found in site 4 of dDISP. It is possible that the cholesterol in site 4 is required for stabilizing the TMs of hDISP; however, dDISP does not require cholesterol to maintain the conformation of dDISP because the cholesterol is not an essential component in the *Drosophila* membrane.

## Materials and Methods

### Protein expression and purification

The constructs of human DISP1 were cloned into pEG BacMam vector with an N-terminal Strep-tag II (WSHPQFEK) and a C-terminal Flag tag (DYKDDDK). The protein was expressed using baculovirus-mediated transduction of mammalian HEK293S GnTI⁻ cells (CRL-3022; ATCC). At 8 h postinfection, the cell culture was supplemented with 10 mM sodium butyrate and transferred to 30°C. After further incubation for 60 h, the cells were disrupted by sonication on ice in buffer A (20 mM Hepes, pH 7.5, 150 mM NaCl), 1 mM PMSF, and 10 µg/ml leupeptin. The sonication was followed by centrifugation at 3,000$g$ for 5 min at 4°C. The resulting supernatant was supplemented with 1% (wt/vol) n-dodecyl-β-D-maltopyranoside (DDM; Anatrace) and 0.1% (wt/vol) cholesteryl hemisuccinate (CHS; Sigma-Aldrich) and incubated for 1 h at 4°C. Insoluble components were removed by centrifugation at 40,000$g$, 4°C, 30 min. The clarified lysate was incubated with anti-FLAG M2 antibody resin (Sigma-Aldrich) for 1 h at 4°C and then loaded onto a disposable gravity column (Bio-Rad). After washing the column with 2 × 10 column volumes (CV) of buffer B (20 mM Hepes, pH 7.5, 400 mM NaCl, 0.02% DDM, and 0.002% CHS), the proteins were eluted with 5 CV of buffer B containing 0.1 mg/ml 3× FLAG peptide (Sigma-Aldrich). The eluted protein was further purified by size-exclusion chromatography

using a Superose 6 Increase 10/300 GL column (GE Healthcare) pre-equilibrated in buffer C (buffer A with 0.06% [wt/vol] digitonin [Sigma-Aldrich] and 0.002% CHS) and 1.5 ml of peak fractions were pooled. To assemble the DISP1*–SHH-N complex, native SHH-N (purchased from R&D Systems, Inc., Cat. no. 8908-SH/CF) was mixed with purified DISP1* at a 1:1.1 molar ratio (DISP1*: SHH-N) and purified by Superose 6 Increase 10/300 GL size-exclusion chromatography in buffer C. The peak fractions (1.5 ml) were collected and concentrated to 5–7 mg/ml for grid preparation. DISP1*–NNN was purified with the same procedure described above.

### Pull-down assay

The wild-type and mutant DISP1 proteins were purified as described above in detergent containing buffer. The HEK293-derived native human SHH-N protein was purchased from R&D Systems, Inc. (Cat. no. 8908-SH/CF). For the pull-down assay, purified DISP1* or DISP1*–NNN protein was immobilized to 20 µl anti-FLAG M2 antibody resin, which was further incubated with native SHH-N for 1 h at 4°C in 200 µl buffer C. Then, the resin was spun down and washed three times with buffer C. The protein complex was eluted with 30 µl buffer C supplemented with 0.3 mg/ml FLAG peptide. 25 µl of the elution was loaded on Bolt 4–12% Bis-Tris Plus Gel (Invitrogen) for detection.

### EM sample preparation and imaging

The freshly purified DISP1*–SHH-N complex was added to Quantifoil R1.2/1.3 400 mesh Au holey-carbon grids (Quantifoil), blotted using a Vitrobot Mark IV (FEI) and frozen in liquid ethane. The grids were imaged in a 300-keV Titan Krios (FEI) with a Gatan K3 Summit direct electron detector (Gatan). Data were collected in super-resolution mode at a pixel size of 0.833 Å with a dose rate of 28 electrons per pixel per second. Images were recorded for 1.5-s exposures in 30 subframes to give a total dose of 62 electrons per Å$^2$.

## Imaging processing and 3D reconstruction

Dark subtracted movies of the DISP1*–SHH-N complexes were normalized by gain reference and the motion correction was performed using MotionCor2 (Zheng et al, 2017). The contrast transfer function was estimated using CTFFIND4 (Rohou & Grigorieff, 2015). To generate the templates for automatic picking, around 2,000 particles of each data set were manually picked and classified by 2D classification in RELION-3 (Zivanov et al, 2018). After auto-picking in RELION-3, the low-quality images and false-positive particles were removed manually. The remaining particles were extracted for subsequent 2D and 3D classification.

A low-resolution cryo-EM map of DISP1* generated from a data set collected on 200 keV Talos Arctica (FEI) was used as the initial model for 3D classification of the Titan Krios data set in RELION-3. The best class from 3D classification was selected for a second round of 2D and 3D classification. The particles of the best two classes from the second-round 3D classification were combined. Then, Bayesian polishing was applied to the particles, followed by the final 3D-refinement with a soft mask and solvent-flattened Fourier shell correlations (FSCs). The final cryo-EM map after RELION-3 refinement was sharpened using postprocess with a B-factor value of $-180 \text{ Å}^2$. The resolution was estimated using "post-processing" with the FSC criteria of 0.143. Local map resolution was estimated by ResMap (Kucukelbir et al, 2014).

## Model construction, refinement and validation

The final maps were B-factor sharpened by "post-processing" in RELION-3 for the model building and refinement. The position of SHH-N in the cryo-EM map was observed by the relatively weak densities in the DISP1*–SHH-N complex (Figs S2 and S3). The structure of *Drosophila* Disp (kindly provided by V Korkhov) was docked into the map as the initial model for DISP1*. The structure model of DISP1* were manually built by COOT (Emsley & Cowtan, 2004), followed by refinement in real space using PHENIX (Adams et al, 2010) and in reciprocal space using Refmac with secondary-structure restraints and stereochemical restraints (Murshudov et al, 1997; Brown et al, 2015). The densities of residues 172–178, 253–294, 306–312, 337–347, 390–414, 442–446, 477–484, 663–687, 771–777, 866–871, 897–902, and 1,145–1,249 were not resolved nor built. Residues 245–252, 304–305, 313–315, 336–338, 415–418, 437–441, 447–451, 473–476, and 485–486 were built with poly-alanine because of limited local resolution. For cross-validations, the final model was refined against one of the half maps from the final 3D-refinement. The resulting model was used to calculate the model versus map FSC curves against the same half map and the other half map, respectively, using the comprehensive validation module in PHENIX (Adams et al, 2010). MolProbity (Chen et al, 2010) was used to validate the geometries of the model. Structure figures were generated using PyMOL (http://www.pymol.org) and Chimera (Pettersen et al, 2004).

## Retroviral infection

Human full-length DISP1 (DISP1FL) and DISP1* with a C-terminal Flag tag were constructed into pMSCV-hygro vectors. Human full-length

SHH cDNA was cloned into pMSCV-puro vector. The viral supernatants were obtained by harvesting media from HEK293T cells that were transiently transfected with pMSCV-hSHH-puro, pMD2.G, and pMDLg/pRRE (Addgene). *Disp1*$^{-/-}$ MEF cells (from Dr. Philip Beachy group) were plated at a density of 20% cells per 6 cm dish, and 24 h later, the cells were transduced with the viral supernatants with 8 $\mu$g/ml polybrene (Sigma-Aldrich). The transduced cells were selected with 3 $\mu$g/ml puromycin (Sigma-Aldrich) and recovered with 0.8 $\mu$g/ml puromycin. *Disp1*$^{-/-}$ MEF cells with stably integrated SHH were transduced with viruses containing an expression cassette for either DISP1FL or DISP1* using the procedure described above followed by the selection with 100 $\mu$g/ml hygromycin B (Sigma-Aldrich). The expression of SHH and DISP1 was confirmed by qRT-PCR.

## HH reporter assays

HEK293 Flp-In T-REX cells with stably integrated Scube2-3×Flag expression cassette (from Dr. Philip Beachy group) were incubated in DMEM-HG (Invitrogen) containing 2% FCS. The expression of Scube2 was induced by 1 $\mu$g/ml doxycycline (Sigma-Aldrich) for 48 h. The Scube2-conditioned medium were filtered with a 0.22-$\mu$m filter (Sigma-Aldrich) and diluted with fresh medium at 1:1 ratio. The SHH-N conditioned medium was prepared by incubation of *Disp1*$^{-/-}$ MEF cells carrying stably integrated SHH and DISPFL/DISP1* expression construct with diluted SCUBE2-conditioned medium for 48 h. The full-length SHH was processed within the cells, which secrete SHH-N into the culture medium. The medium was filtered and diluted with fresh DMEM with 0.5% newborn calf serum at 1:1 ratio. SHH-Light II cells with stably expressed firefly luciferase and *Renilla* luciferase were seeded at 100,000 per well of 24-well plates in DMEM with 0.5% newborn calf serum 1 d before experiment, followed by incubation with diluted SHH-N conditioned medium for 28 h. The luciferase activity was measured using the Dual-Luciferase Reporter Assay System (Promega). The expression of the processed SHH-N was detected by Western blotting using anti-SHH-N antibody (sc-365112; Santa Cruz Biotechnology). Each assay was reproduced at least three times, and data were analyzed using Excel (Microsoft). Graphs were generated by Prism 8 (GraphPad).

## Immunoblot analysis

*Disp1*$^{-/-}$ MEF cells carrying stably integrated SHH and DISPFL/DISP1* expression construct were incubated as described above. The cells were washed with PBS for three times, then incubated with 1 mM EDTA-containing PBS for 10 min at 37°C. The cells were suspended and then centrifuged at 3,000$g$ for 10 min at 4°C. The resulting pellets were resuspended in buffer A (20 mM Hepes, pH 7.5, 150 mM NaCl), 1 mM PMSF, and 10 $\mu$g/ml leupeptin. The resuspended cells were supplemented with 1% DDM and incubated at 4°C for at least 1 h. Followed by high-speed centrifugation, the resulting supernatant was incubated with 4× Bolt LDS Sample Buffer (Invitrogen).

The culture medium from above cells were harvested and filtered to remove any detached cells. The resulting supernatant was concentrated to small volume using Amicon Ultra-15 Centrifugal Filter Units, 10-kD cutoff (Millipore). The concentrated supernatant

was mixed with 4× Bolt LDS Sample Buffer. After electrophoresis, the proteins from above samples were transferred to nitrocellulose filters, which were then incubated with either anti-SHH-N antibody (sc-365112; Santa Cruz Biotechnology) or anti-FLAG M2 antibody (F3165; Sigma-Aldrich) at 4°C overnight, followed by the incubation of HRP-linked antimouse IgG (#7076; Cell Signaling Technology) at room temperature for 1 h. Bound antibodies were visualized by a SuperSignal West Pico PLUS Chemiluminescent Substrate Kit (Thermo Fisher Scientific). The images were scanned and analyzed using an Odyssey Fc Imaging System (LI-COR Biosciences). Similar results were obtained in three biologically independent experiments.

### Transduction verification by qRT-PCR

RNA was isolated from cell cultures using TRIzol (Thermo Fisher Scientific), according to manufacturer's instructions. cDNA was synthesized using M-MLV Reverse Transcriptase (Invitrogen) and Random Primers (Invitrogen). Relative mRNA levels were determined by qRT-PCR using SYBR Green PCR Master Mix (Applied Biosystems). Values were normalized to Rps18 levels using the ΔΔ-Ct method.

The primer sequences used are as follows:

*DISP1* Forward 5′-TTTTAACATCGCCAGCCCAGC-3′
*DISP1* Reverse 5′-AGCAGCTGGTGAAGTCCTGTT-3′
*Rps18* Forward 5′-CATGCAAACCCACGACAGTA-3′
*Rps18* Reverse 5′-CCTCACGCAGCTTGTTGTCTA-3′.

## Data Availability

The data that support the findings of this study are available from the corresponding author upon request. The 3D cryo-EM density map has been deposited in the Electron Microscopy Data Bank under the accession number EMD-22144. Atomic coordinate for the atomic model has been deposited in the Protein Data Bank under the accession number 6XE6. Correspondence and requests for materials should be addressed to X Li.

## Supplementary Information

## Acknowledgements

We thank D Stoddard at the University of Texas (UT) Southwestern Medical Center Cryo-EM Facility (funded in part by the CPRIT Core Facility Support Award RP170644) for assistance in data collection. We thank P Beachy for sharing the *Disp1*$^{-/-}$ MEFs and stable cell line for SCUBE2 expression; V Korkhov for structure coordinates of dDISP1 prior to publication; and R DeBose-Boyd, E Debler, L Friedburg, T Long, and X Qi for manuscript preparation. This work was supported by the Endowed Scholars Program in Medical Science of UT Southwestern Medical Center, National Institutes of Health (NIH) grant P01 HL020948, NIH grant R01 GM135343 (to X Li). X Li is a Damon Runyon-Rachleff Innovator supported by the Damon Runyon Cancer Research Foundation (DRR-53-19) and a Rita C. and William P. Clements Jr. Scholar in Biomedical Research at UT Southwestern Medical Center.

### Author Contributions

H Chen: data curation, formal analysis, validation, investigation, and writing—review and editing.
Y Liu: formal analysis, validation, investigation, and writing—review and editing.
X Li: conceptualization, formal analysis, supervision, funding acquisition, and writing—original draft, review, and editing.

### Conflict of Interest Statement

The authors declare that they have no conflict of interest.

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
