## [Reviewer comments · Life Science Alliance]

Life Science Alliance

Structure of Human Dispatched-1 Provides Insights into Hedgehog Ligand Biogenesis

Hongwen Chen, Yang Liu, and Xiaochun Li
DOI: 10.26508/lsa/202000776

Corresponding author(s): Xiaochun Li, UT Southwestern Medical Center

Review Timeline:

Submission Date:	2020-05-13
Editorial Decision:	2020-06-02
Revision Received:	2020-06-17
Editorial Decision:	2020-06-22
Revision Received:	2020-06-23
Accepted:	2020-06-24

Scientific Editor: Reilly Lorenz

Transaction Report:

June 2, 2020

Re: Life Science Alliance manuscript #LSA-2020-00776-T

Xiaochun Li
UT Southwestern Medical Center

Dear Dr. Li,

Thank you for submitting your manuscript entitled "Structure of Human Dispatched-1 Provides Insights into Hedgehog Ligand Biogenesis" to Life Science Alliance. The manuscript was assessed by expert reviewers, whose comments are appended to this letter.

The referees appreciate the findings and have noted a few minor revisions that need to be sorted out. You can use the link below to upload the revised version.

Thank you for this interesting contribution to Life Science Alliance. We are looking forward to receiving your revised manuscript.

Sincerely,

Reilly Lorenz
Editorial Office Life Science Alliance
Meyerhofstr. 1
69117 Heidelberg, Germany
t +49 6221 8891 414
e contact@life-science-alliance.org

B. MANUSCRIPT ORGANIZATION AND FORMATTING:

Reviewer #1 (Comments to the Authors (Required)):

This is a high quality report of the cryoEM structure of Dispatched, a protein needed for the release of Hedgehog from cells. The authors show that unlike the related PTC and NPC proteins, this protein has an open extracellular domain that binds its cholesterol and palmitoylated HH ligand. The story should be published without delay given a recent publication of the related Drosophila protein after consideration of the following minor points.

1. Please give more description to what is shown in the Fig. legends (1B, explain); 1C, state which luciferase reporter is used.
2. Fig. 3A is hard to decipher. Perhaps showing the related proteins, side by side would be much easier for the reader to understand.

3. It was not clear to this reader why HH would cause Disp1 to form a dimer?

4. cholesterol is seen throughout the transmembrane domains. The authors should at least discuss the possibility that Disp is a cholesterol transporter like its relatives? Even if they don't favor such a model, it is important to consider.

Otherwise, well done! -Suzanne Pfeffer, Signed review

Reviewer #2 (Comments to the Authors (Required)):

The manuscript by Chen and colleagues reports a 4.5 Å resolution cryo-EM structure of human Dispatched (DISP1), which is critical for hedgehog (Hh) signaling. The authors compare structures of DISP1 and other related Resistance-nodulation-division (RND) transporters including PTCH1 and NPC1. In particular, the extracellular domains in DISP1 show large conformational differences in comparison to those in PTCH1 and NPC1. This work improves our understanding of how DISP1 might mediate Hh release.

Recently, a much higher resolution (3.2 Å) cryo-EM structure of the *D. melanogaster* Dispatched and structure of Dispatched in complex with a modified Hh ligand were reported (Sci. Adv. 2020; 6 : eaay7928). The current manuscript by Chen and colleagues reports a much lower resolution model. At 4.5 Å resolution, it is challenging to register the amino acid sequence into the density map as a result of lack of sufficient side chain densities. Therefore, interpretation of the structural model requires extra caution. The authors can increase the accuracy of the human DISP1 structural model by taking advantage of the much higher resolution structures of *D. melanogaster* Dispatched. For instance, sequence alignment of human DISP1 and *D. melanogaster* Dispatched, together with the higher resolution structure of *D. melanogaster* Dispatched, can improve modeling of the human DISP1 structure. In addition, structural comparison of human DISP1 and *D. melanogaster* Dispatched and thorough analysis would improve the manuscript by making it more relevant in the field. This is the major concern and it should be thoroughly addressed.

Minor points:

Cryo-EM map should be provided in the main figures.

At this low resolution, the authors should be very cautious about assigning sterol densities and the statements should be toned down.

In Fig. S1, the DISP1-HH-N complex migrates much slower than DISP1 alone. What is the explanation?

Reviewer #3 (Comments to the Authors (Required)):

Human Dispatched-1 (DISP1) is the transmembrane protein responsible for secretion of Hedgehog (HH). HH signalling is important for metazoan development but the mechanism of HH release from the cell surface by DISP1 is unclear. To better understand this, the authors solve a 4.5 Å resolution structure of a truncated version of DISP1 (DISP1*) using cryo-electron microscopy (cryo-EM). In the cryo-EM structure, the extracellular domains (ECDs) of DISP1* are positioned further apart than

seen in closely related proteins, PTCH1 and NPC1. The authors also observe five cholesterol-like molecules adjacent to transmembrane helices in DISP1. These features are clearly visible in the cryo-EM density. In addition to the structure of DISP1* alone, the authors determined a ~8 Å structure of DISP1* bound to the HH ligand using the same dataset, showing the likely binding surface of HH to DISP1.

Overall, the data fits well with the claims made and the conclusions are not overstated. This work shows the mechanism of ligand binding and release from DISP1 is likely different from PTCH1 and NPC1. It is complementary to a recently published structure of *Drosophila* DISP, which contains lower occupancy of a cholesterol-like molecule in the transmembrane region and clearer HH ligand density in the EM map. These differences may have functional relevance which could be investigated in future. In the current manuscript, some structural features are not clearly visible in the Figures and the meaning of some parts of the discussion are not clear. I have outlined points for improvement below.

1) The authors state: "This structural analysis together with the findings in PTCH1 suggest that the three Asp residues in DISP1 may provide the energy source for the conformational changes of DISP1 facilitating the release of HH-N." The mechanism proposed is not clear. The authors should be more specific about why the greater distance between the three Aspartate residues could be an energy source for HH-N release in DISP1. What conformational changes are thought to occur during HH-N binding and release? How is this related to the presence of cholesterol-like molecules in the TMs?

2) The speculation that Furin cleavage is responsible for the open conformation of the ECDs in DISP1* is not discussed in detail. Are residues equivalent to DISP1 253-292 in NPC1/PTCH1 ordered? What is extended length of peptide between these residues - could the distance between ECDs be accommodated without Furin cleavage?

3) The authors show the truncated construct used for structure determination (DISP1*) is able to bind and secrete SHH-N and so is a functional construct. However, Figure S2 shows DISP1* is expressed at a level ~3 times higher than DISP1FL. Figure 1C shows similar levels of SHH-N secretion from these cells. This suggests DISP1* has lower activity than DISP1FL. This should be made clear in the text.

4) The text states that the DISP1* had "ideal behavior in solution" but Figure S1A shows aggregation and presence of a higher molecular weight species when run on gel-filtration. The fractions pooled and re-run after SSH-N addition should be indicated. Were the higher molecular weight species contaminating proteins or oligomers of Disp1*?

5) The electron density for the DISP1* structure is only shown in a small panel in Figure S3. This should be moved to Figure 2 so that the quality of electron density can be easily assessed by readers.

6) The position of structural features described in the text is not clear in Figure 2. Cartoons and expanded views of parts of the structure should be added to clearly show the location of missing density between residues 253-292, the α 1 and α 1* helices and the TM helices mentioned in the text.

7) The comparison of the DISP1* structure to PTCH1 and NPC1 in Figure 3 are not clear. Surface representation or thicker cartoon representation with different colors could be used to show the

conformational changes clearly.

8) The authors refer to the SHH-N and HH-N at different points in the manuscript (both DISP1*-HH-N and DISP1*-SHH-N complexes are referred to). This should be made consistent to avoid confusion.

Reviewer #1 (Comments to the Authors (Required)):

This is a high quality report of the cryoEM structure of Dispatched, a protein needed for the release of Hedgehog from cells. The authors show that unlike the related PTC and NPC proteins, this protein has an open extracellular domain that binds its cholesterol and palmitoylated HH ligand. The story should be published without delay given a recent publication of the related *Drosophila* protein after consideration of the following minor points.

1. Please give more description to what is shown in the Fig. legends (1B, explain); 1C, state which luciferase reporter is used.

Response: Fig.1B legend has been modified as follow: “Experimental scheme to measure SHH release capacity. HEK293 Flp-In T-REX cells were stably transfected with an inducible 3X Flag-tagged Scube2. Cells were induced and the supernatant containing Scube2 conditioned medium was collected, filtered and mixed with fresh medium at a 1:1 ratio and finally transferred to DISP1^{-/-} or DISP1* and DISP^{FL} rescued MEF cells stably expressing FL-SHH. After 48h of incubation, the SHH-N conditioned medium was harvested and mixed with fresh medium at 1:1 ratio to incubate with SHH-Light II cells.”

The Fig.1C legend has been changed to: “SHH-N release in DISP1^{-/-} MEF cells that were transduced with both SHH and empty vector (DISP1^{KO}), Full-length DISP1 (DISP1^{FL}) or DISP1* was checked via dual luciferase assay where the SHH-Light II cells stably express firefly luciferase with a 8X-Gli promoter and Renilla luciferase with a constitutive promoter . Data are mean ± s.d. (n = 8 biologically independent experiments).”

2. Fig. 3A is hard to decipher. Perhaps showing the related proteins, side by side would be much easier for the reader to understand.

Response: Point accepted. We have changed the figure by putting the overall structures of DISP1*, PTCH1 and NPC1 side by side for comparison and label each domain directly for clarity.

3. It was not clear to this reader why HH would cause Disp1 to form a dimer?

Response: We didn't observe any experimental evidence that shows HH would cause Disp1 to form a dimer. So, we have rewritten the sentence to indicate that it's unclear whether SHH-N can also bind two DISP1* in a similar manner as seen in PTCH1-SHH-N 2:1 complex.

“...interfaces to form a 2:1 complex (Qi et al. 2018). However, it remains unknown whether SHH-N can also bind two DISP1* molecules with different surface residues to form an oligomer.” (Pages 9, lines 161-162)

4. cholesterol is seen throughout the transmembrane domains. The authors should at least discuss the possibility that Disp is a cholesterol transporter like its relatives? Even if they don't favor

such a model, it is important to consider.

Response: We agree with this reviewer. The possibility has been discussed as following.
“...the corresponding residues in PTCH1 or NPC1 (Fig. 4D and E). This structural analysis suggests that the three Asp residues in DISP1 may constitute a core ion-conducting circuit to use a different ion gradient to provide the energy source for transporter-like activity. However, the relation between transporter-like activity and the release of HH-N is still unclear. One appealing hypothesis is that the transmembrane ion gradient drives DISP1 to transport cholesterol or its derivatives between inner and outer leaflets, which regulates the release of SHH-N. Alternatively, the cholesterol-like molecule in site 4 may increase...” (Page 10-11, lines 205-211)

Otherwise, well done! -Suzanne Pfeffer, Signed review

The authors thank this referee for her time and constructive comments.

Reviewer #2 (Comments to the Authors (Required)):

The manuscript by Chen and colleagues reports a 4.5 Å resolution cryo-EM structure of human Dispatched (DISP1), which is critical for hedgehog (Hh) signaling. The authors compare structures of DISP1 and other related Resistance-nodulation-division (RND) transporters including PTCH1 and NPC1. In particular, the extracellular domains in DISP1 show large conformational differences in comparison to those in PTCH1 and NPC1. This work improves our understanding of how DISP1 might mediate Hh release.

Recently, a much higher resolution (3.2 Å) cryo-EM structure of the *D. melanogaster* Dispatched and structure of Dispatched in complex with a modified Hh ligand were reported (Sci. Adv. 2020; 6 : eaay7928). The current manuscript by Chen and colleagues reports a much lower resolution model. At 4.5 Å resolution, it is challenging to register the amino acid sequence into the density map as a result of lack of sufficient side chain densities. Therefore, interpretation of the structural model requires extra caution. The authors can increase the accuracy of the human DISP1 structural model by taking advantage of the much higher resolution structures of *D. melanogaster* Dispatched. For instance, sequence alignment of human DISP1 and *D. melanogaster* Dispatched, together with the higher resolution structure of *D. melanogaster* Dispatched, can improve modeling of the human DISP1 structure. In addition, structural comparison of human DISP1 and *D. melanogaster* Dispatched and thorough analysis would improve the manuscript by making it more relevant in the field. This is the major concern and it should be thoroughly addressed.

Response: Point accepted. Before we submitted the manuscript, we have used the structure of *D. melanogaster* Dispatched to facilitate the model building. Therefore, we believe that the register assignment of human DISP1 is correct. Also, the structural comparison has been performed and discussed in the revision (Fig. S7 and Pages 11-12 lines 226-234).

Minor points:

Cryo-EM map should be provided in the main figures.

Response: Following this reviewer's suggestion, we revised Figure 2 in the current manuscript.

At this low resolution, the authors should be very cautious about assigning sterol densities and the statements should be toned down.

Response: Point accepted. We assigned the sterol molecules based on two rationales: 1) the density shapes and the hydrophobic environment that is created protential binding residues; 2) the similar binding sites that were found in PTCH1. We have toned down the statements of the putative sterol molecules.

In Fig. S1, the DISP1-HH-N complex migrates much slower than DISP1 alone. What is the explanation?

Response: The two ECDs of DISP1 adopt an open conformation and leave a cleft between them. Upon binding to SHH-N, the two ECDs of DISP1 may form more compact dimensions, therefore the complex migrates much slower than DISP1 alone.

The authors thank this referee for his/her time and constructive comments.

Reviewer #3 (Comments to the Authors (Required)):

Human Dispatched-1 (DISP1) is the transmembrane protein responsible for secretion of Hedgehog (HH). HH signalling is important for metazoan development but the mechanism of HH release from the cell surface by DISP1 is unclear. To better understand this, the authors solve a 4.5 Å resolution structure of a truncated version of DISP1 (DISP1*) using cryo-electron microscopy (cryo-EM). In the cryo-EM structure, the extracellular domains (ECDs) of DISP1* are positioned further apart than seen in closely related proteins, PTCH1 and NPC1. The authors also observe five cholesterol-like molecules adjacent to transmembrane helices in DISP1. These features are clearly visible in the cryo-EM density. In addition to the structure of DISP1* alone, the authors determined a ~8 Å structure of DISP1* bound to the HH ligand using the same dataset, showing the likely binding surface of HH to DISP1.

Overall, the data fits well with the claims made and the conclusions are not overstated. This work shows the mechanism of ligand binding and release from DISP1 is likely different from PTCH1 and NPC1. It is complementary to a recently published structure of Drosophila DISP, which contains lower occupancy of a cholesterol-like molecule in the transmembrane region and clearer HH ligand density in the EM map. These differences may have functional relevance which could be investigated in future. In the current manuscript, some structural features are not clearly visible in the Figures and the meaning of some parts of the discussion are not clear. I have outlined points for improvement below.

1) The authors state: "This structural analysis together with the findings in PTCH1 suggest that the three Asp residues in DISP1 may provide the energy source for the conformational changes of DISP1 facilitating the release of HH-N." The mechanism proposed is not clear. The authors should be more specific about why the greater distance between the three Aspartate residues could be an energy source for HH-N release in DISP1. What conformational changes are thought to occur during HH-N binding and release? How is this related to the presence of cholesterol-like molecules in the TMs?

Response: It has been well established that most RND superfamily proteins function as H⁺-driven antiporters (Tseng et al. 1999; Nikaïdo 2018), but some other RND transporters in halophilic organisms have been identified to be Na⁺-driven (Ishii et al. 2015). A recent study suggests that PTCH1 specifically requires the transmembrane Na⁺ gradient to power PTCH1 transporter-like activity in Smoothed regulation (Myers et al. 2017). The greater distance between the three Aspartate residues may accommodate a different ion to drive the putative transporter-like activity of DISP1. However, the relation between transporter-like activity and the release of HH-N is still unclear and beyond the focus of the current manuscript.

2) The speculation that Furin cleavage is responsible for the open conformation of the ECDs in DISP1* is not discussed in detail. Are residues equivalent to DISP1 253-292 in NPC1/PTCH1 ordered? What is extended length of peptide between these residues - could the distance between ECDs be accommodated without Furin cleavage?

Response: The residues equivalent to DISP1 253-292 are absent in both NPC1 and PTCH1. Theoretically, the maximum length of a 40-aa peptide is about 120 Å, which is more than twice

the distance between ECDs. It seems unlikely that the loop between residues 253 to 292 would restrict the ECDs to form a closed conformation. Moreover, Disp1 cleavage by Furin is not required to interact with HH-N (Stewart et al. 2018). So, we will not claim this point in the revised manuscript. As such, we have modified the section as the following.

“On the other hand, the loop between residues 253 to 292, which is absent in both PTCH1 and NPC1, is disordered in the map (Fig. 2A and C), indicating that the flexibility of this large loop permits it to be accessible to Furin and the cleavage by Furin may release this internal loop to facilitate the endosomal trafficking of DISP1 (Stewart et al. 2018).” (Pages 7-8, lines 135-139)

3) The authors show the truncated construct used for structure determination (DISP1*) is able to bind and secrete SHH-N and so is a functional construct. However, Figure S2 shows DISP1* is expressed at a level ~3 times higher than DISP1^{FL}. Figure 1C shows similar levels of SHH-N secretion from these cells. This suggests DISP1* has lower activity than DISP1^{FL}. This should be made clear in the text.

Response: Biochemical and cell biological analyses have shown mammalian Disp1 assemblies into functional trimers through its C-terminal domain (CTD), which is required for its localization to basolateral cell surface. Deletion of the entire CTD causes a mild reduction of SHH-N secretion and an increase in Shh-N retention (Etheridge et al. 2010). However, the lack of conservation across species and dramatic flexibility of Disp1 CTD hinders recombinant expression and purification in large quantity for further structural study. Furthermore, sequence alignment shows that *Drosophila* Disp1 lacks the sequences corresponding to the CTD of mammalian Disp1 (Ma et al. 2002). So, we speculate that the lower activity of DISP1* is attributed to its partially mislocalization rather than loss of SHH-N secretion activity per se.

Following this reviewer’s suggestion, we have rewritten the sentences as following: “...processed SHH-N in both lines. Furthermore, the secreted SHH-N functionally activated SHH-Light II cells, an NIH 3T3-derived line stably transfected with Gli-dependent firefly luciferase and constitutive *Renilla* luciferase. Notably, RT-qPCR (Fig. S2) showed an approximately 3-fold higher expression of DISP1* than DISP1^{FL}. Despite this higher expression the overall amount of secreted SHH-N was relatively similar. This is consistent with a previous report showing that DISP1 lacking the C-terminal domain results in a mild reduction of SHH secretion (Etheridge et al. 2010). Even so, these results show that DISP1* is functional and suitable for further studies.” (Pages 5-6, lines 90-98)

4) The text states that the DISP1* had "ideal behavior in solution" but Figure S1A shows aggregation and presence of a higher molecular weight species when run on gel-filtration. The fractions pooled and re-run after SSH-N addition should be indicated. Were the higher molecular weight species contaminating proteins or oligomers of Disp1*?

Response: Indeed, in each purification we observed the higher molecular weight species, including a peak at the position of void volume and a shoulder ahead of the main peak. These fractions were subjected to gel electrophoresis and Coomassie staining showed that these higher molecular weight species were oligomers of DISP1*.

Following this reviewer's suggestion, we modified the "Protein expression and purification" part of Methods section to indicate the fractions pooled after each gel-filtration.

5) The electron density for the DISP1* structure is only shown in a small panel in Figure S3. This should be moved to Figure 2 so that the quality of electron density can be easily assessed by readers.

Response: Following this reviewer's suggestion, we have added a panel in Figure 2 to show the electron density for the DISP1* structure.

6) The position of structural features described in the text is not clear in Figure 2. Cartoons and expanded views of parts of the structure should be added to clearly show the location of missing density between residues 253-292, the $\alpha 1$ and $\alpha 1^*$ helices and the TM helices mentioned in the text.

Response: Point accepted. We have changed the Figure 2 according to the suggestions of this reviewer and re-written the text to describe the figure.

7) The comparison of the DISP1* structure to PTCH1 and NPC1 in Figure 3 are not clear. Surface representation or thicker cartoon representation with different colors could be used to show the conformational changes clearly.

Response: Point accepted. We have changed the figures according to this suggestion.

8) The authors refer to the SHH-N and HH-N at different points in the manuscript (both DISP1*-HH-N and DISP1*-SHH-N complexes are referred to). This should be made consistent to avoid confusion.

Response: Point accepted. We used "SHH-N" for the complex assembly, and "HH-N" for the signaling molecule across species in the text.

The authors thank this referee for his/her time and constructive comments.

June 22, 2020

RE: Life Science Alliance Manuscript #LSA-2020-00776-TR

Dr. Xiaochun Li
UT Southwestern Medical Center
5323 Harry Hines Blvd.
L5.162
Dallas, Texas 75390

Dear Dr. Li,

Thank you for submitting your revised manuscript entitled "Structure of Human Dispatched-1 Provides Insights into Hedgehog Ligand Biogenesis". We would be happy to publish your paper in Life Science Alliance pending final revisions necessary to meet our formatting guidelines.

-In your abstract, We suggest to change the wording from
"Its ECDs reveal an open state, opposite of its structural homologues..."

TO
"Its ECDs reveal an open state, different from (OR: in contrast to) its structural homologues..."

- please consult our Manuscript Preparation Author Guidelines and follow the instructions of how the sections of your manuscript should be divided
- please add a scale bar to figure S3A and define its size in the legend
- please fill in the conflict of interest statement in our system
- please upload your supplementary figure files as singular files
- please add a callout to Table S1 and provide the tables in excel format
- please add the supp. figure legends to the main manuscript
- please provide your manuscript in docx format

A. FINAL FILES:

B. MANUSCRIPT ORGANIZATION AND FORMATTING:

Sincerely,

Reilly Lorenz
Editorial Office Life Science Alliance
Meyerhofstr. 1
69117 Heidelberg, Germany

t +49 6221 8891 414
e contact@life-science-alliance.org
www.life-science-alliance.org

June 24, 2020

RE: Life Science Alliance Manuscript #LSA-2020-00776-TRR

Dr. Xiaochun Li
UT Southwestern Medical Center
5323 Harry Hines Blvd.
L5.162
Dallas, Texas 75390

Dear Dr. Li,

Thank you for submitting your Research Article entitled "Structure of Human Dispatched-1 Provides Insights into Hedgehog Ligand Biogenesis". It is a pleasure to let you know that your manuscript is now accepted for publication in Life Science Alliance. Congratulations on this interesting work.

*****IMPORTANT:** If you will be unreachable at any time, please provide us with the email address of an alternate author. Failure to respond to routine queries may lead to unavoidable delays in publication.*******

DISTRIBUTION OF MATERIALS:

Again, congratulations on a very nice paper. I hope you found the review process to be constructive and are pleased with how the manuscript was handled editorially. We look forward to future exciting submissions from your lab.

Sincerely,

Reilly Lorenz
Editorial Office Life Science Alliance
Meyerhofstr. 1
69117 Heidelberg, Germany
t +49 6221 8891 414
e contact@life-science-alliance.org
www.life-science-alliance.org